# Analysis of Clinical Parameters, Drug Consumption and Use of Health Resources in a Southern European Population with Alcohol Abuse Disorder during COVID-19 Pandemic

**DOI:** 10.3390/ijerph19031358

**Published:** 2022-01-26

**Authors:** Ana Lear-Claveras, Beatriz González-Álvarez, Sabela Couso-Viana, Ana Clavería, Bárbara Oliván-Blázquez

**Affiliations:** 1Aragonese Research Group in Primary Care (Grupo Aragonés de Investigación en Atención Primaria/GAIAP), Aragon Health Research Institute, 50015 Zaragoza, Spain; analearc@gmail.com (A.L.-C.); bolivan@unizar.es (B.O.-B.); 2Biocomputing Department, Instituto Aragonés de Ciencias de la Salud, 50009 Zaragoza, Spain; bgonzalez.iacs@aragon.es; 3I-Saúde Group, South Galicia Health Research Institute, 36201 Vigo, Spain; sabela.couso@iisgaliciasur.es; 4Vigo Health Area, SERGAS, 36201 Vigo, Spain; 5Network for Research on Chronicity, Primary Care and Health Promotion (RICAPPS), 08007 Barcelona, Spain; 6Department of Psychology and Sociology, University of Zaragoza, 50009 Zaragoza, Spain

**Keywords:** COVID-19, alcohol use disorder, lockdown, primary care, lifestyle, health resources

## Abstract

The disruption in healthcare attention to people with alcohol dependence, along with psychological decompensation as a consequence of lockdown derived from the COVID-19 pandemic could have a negative impact on people who suffer from alcohol abuse disorder. Observational real world data pre-post study included 9966 men aged >16 years registered as having the diagnosis of alcohol abuse disorder in the electronic medical records (EMR) of the Aragon Regional Health Service (Spain). Clinical (Glutamate-oxaloacetate -GOT-, Glutamate pyruvate -GPT-, creatinine, glomerular filtration, systolic blood pressure -SBP-, diastolic blood pressure -DBP-, total cholesterol, LDL, HDL, triglycerides, and body mass index -BMI-), pharmacological (dose per inhabitant per day, DHD, of drugs used in addictive disorders, benzodiazepines and antidepressants) and health resource use variables (primary and specialized care) were considered. A Student’s *t*-test for matched samples was performed to analyze the changes in clinical variables between alcohol abuse disorder patients with and without COVID-19. Only creatinine and LDL showed a significant but clinically irrelevant change six months after the end of the strict lockdown. The total number of DHDs for all drugs included in the study (except for benzodiazepines), decreased. In the same way, the use of health services by these patients also decreased. The impact of COVID-19 among this group of patients has been moderate. The reorganization of health and social services after the declaration of the state of alarm in our country made possible the maintenance of care for this vulnerable population.

## 1. Introduction

The COVID-19 outbreak caused an unprecedented public health crisis around the world [1]. The declaration of the coronavirus disease as a pandemic in March 2020, and its dramatic development thereafter, exposed the world population [2], especially at the beginning of the pandemic, to stressful situations [3]. In an effort to contain the spread of this new SARS-CoV-2 virus, the governments of many countries established restrictive measures based on limiting mobility. In the case of Spain, the government declared a state of national emergency on 15 March 2020, forcing citizens to confine themselves to their homes until 3 May [4]. These measures, which were required to slow the spread of the virus, seem to have had a significant impact on the physical and mental health of the population [5].

Although many studies have shown an increase in the levels of stress, anxiety and depression during the months of lockdown among the general population [6,7,8,9,10,11], few studies have evaluated the effect of this pandemic and the consequent lockdown on alcohol consumption patterns, despite the fact that the consumption of this substance could have been used to reduce the intensity of negative feelings caused by home lockdown [12].

It is estimated that in the world there are 237 million men and 46 million women who suffer from alcohol use disorder. In 2016, the harmful use of this substance caused more than 3 million deaths worldwide (three-quarters of these, in men), representing 5% of the global burden of disease [13]. Globally, that same year, alcohol consumption was the seventh highest risk factor for premature death and disability, and was the main risk factor among the population aged 15 to 49 years [14]. Among young men in this age group, alcohol abuse is also the leading cause of disability in our country [15].

Studies on the impact of the pandemic and lockdown on the consumption patterns of this substance in the general population show variability in their results. Some studies report a decrease in consumption during the months of lockdown [16,17], while other studies report an increase in consumption [18,19]. Those who increased their alcohol intake during lockdown experienced higher levels of stress, anxiety or depression than those who maintained their habitual consumption or reduced it [16,18,20,21]; likewise, this increase was associated more frequently with individuals who declared themselves addicted to alcohol before the pandemic [17]. Some studies suggested that the psychological decompensation resulting from lockdown away from a socio-affective network could cause an increase in alcohol consumption and in the number of relapses among those who had alcohol consumption disorders before the pandemic [22,23,24].

Before the pandemic, primary care (PC) services attended to most of the demands related to alcohol problems, therefore, a significant number of visits in these centers were motivated by this problem and by the pathologies related to alcohol consumption [15]. The pandemic, especially at the beginning, forced individuals to modify the functions of the PC teams, among which were the detection and early diagnosis of alcohol abuse disorder, the performance of motivational interventions and the referral to specialized treatment centers.

Changes in daily routines and the interruptions of care services for people with alcohol abuse disorder could have a particularly negative impact on the health status of this group [13,22]. Without a structured routine of non-alcohol related activities and without behavioral therapies, people with alcohol abuse disorders could more easily succumb to drinking during lockdown [22].

This increase in alcohol consumption during the pandemic could in turn worsen their health. Enough evidence has shown the existence of a dose-dependent effect between chronic alcohol consumption and viral infections (hepatitis C, HIV); thus, people with alcohol abuse disorders could have a higher risk of contracting COVID-19 [25]. Among this group, consumption of other substances (tobacco and other drugs) [26,27] and comorbidity with chronic (cancer, cardiovascular disease, liver disease, pancreatitis or diabetes) [28] and psychiatric pathologies (depression, disorder of generalized anxiety or bipolar disorder) is frequent [29], which could make them especially susceptible to a worse prognosis of infection [30].

Despite the fact that various studies have analyzed the impact of the pandemic and lockdown on the population with alcohol dependence during the peak of the pandemic, fewer articles have studied the impact of both in the following months after the end of the lockdown. The aim of this study is to analyze the changes in the clinical parameters, the consumption of drugs for the treatment of this addiction and the use of health resources between the six months before the start of the lockdown and the six months after its end in men diagnosed with alcohol abuse disorder in an autonomous community in northern Spain; we also aim to compare the variations in clinical parameters between patients with alcohol dependence not infected and those who were infected by COVID-19.

## 2. Materials and Methods

### 2.1. Design and Study Population

Observational real world data were collected pre-post study of men over 16 years of age in the Autonomous Community of Aragon (Spain), diagnosed with alcohol abuse disorder in their electronic medical records (EMR) according to the criteria of the International Classification of Primary Care (ICPC-2): code P15 [31]. This disorder is characterized by the loss of control over drinking, cause in the individual the compulsion to drink alcohol continuously or periodically in order to experience psychic effects or avoid the discomfort caused by its absence [32].

From the EMR, information was collected for each individual in the six months prior to the start of the lockdown (14 September 2019–15 March 2020) and in the six months after its end (3 May 2020–11 April 2020). The records collected during the months of strict lockdown were not considered because these were very scarce.

Finally, to find out which clinical parameters might be present more frequently in individuals with and without the infection we also investigated the differences in clinical parameters in the six months prior to the start of lockdown and in the six months after its ends, among individuals with a diagnosis of alcohol abuse who did not contract COVID-19 during the period of study and those who contracted the infection.

### 2.2. Data Sources

This study is based on data from the electronic medical records of the PC services of Aragon.

The implementation of EMR was completed throughout the Aragon health system in 2011. This record, shared by all the professionals that are part of the health system, contains the data generated throughout the care process (PC and hospital care) by the patients covered under the National Health System.

### 2.3. Variables

The sociodemographic variables included in this study were: sex, age, pharmaceutical service and basic health area. The number of deaths among the population under study for each of the measurement periods was also collected; as well as chronic comorbidities with prevalences greater than 5% [33] (arrhythmias, heart failure, ischemic heart disease, hypertension, dyslipidemia, obesity, overweight, disease in veins and arteries, cerebrovascular disease, diabetes, chronic bronchitis, COPD, asthma, disease kidney, hypo and hyperthyroidism anemia, neoplasia, hearing loss, cataracts, glaucoma, osteoarthritis, osteoporosis, dorsopathy, smoking, insomnia, chronic anxiety and depression, attempted suicide, and dementia).

Regarding the clinical and analytical parameters related to chronic alcohol consumption, the following were included: glutamic oxaloacetic transaminase (GOT), glutamic pyruvic transaminase (GPT), blood creatinine, glomerular filtration, systolic (SBP) and diastolic blood pressure (DBP), total cholesterol, low-density lipoprotein (LDL), high-density lipoprotein (HDL), triglycerides, and body mass index (BMI).

Changes in pharmacological treatment were assessed through variations in the total number of defined daily doses per 1000 inhabitants per day (DHD) dispensed in the pharmacy. DHDs were calculated from the defined daily dose (DDD) stipulated by the World Health Organization (WHO), according to the following formula:(1)$$\mathrm{DHD}=\frac{\mathrm{Registered}\;\mathrm{consumption}\;\mathrm{of}\;\mathrm{the}\;\mathrm{active}\;\mathrm{ingredient}*1000\;\mathrm{inhabitants}}{\mathrm{Standard}\;\mathrm{DDD}*n\underline{{}^{\circ}}\mathrm{inhabitants}/\mathrm{period}*365\;\mathrm{days}}$$

Taking into account the anatomical, therapeutic, chemical (ATC) classification system, the codes of the drugs of choice for the treatment of this pathology were analyzed according to the Spanish Society of Family and Community Medicine (Sociedad Española de Medicina de Familia y Comunitaria—semFYC) [34]: N07 (drugs used in additive disorders), N05 (benzodiazepines) and N06 (antidepressants).

Lastly, the use of health resources by these patients was assessed with the use of PC services (number of ordinary or continuous care visits at the health center or at home by the nurse or the general practitioner, and number of visits to other professionals in the health center) and with the use of hospital services (number of visits in outpatient care, number of diagnostic tests performed, number of visits to the emergency service, and number of hospitalizations) for each of the periods.

### 2.4. Statistical Analysis

The sample size allowed the use of parametric methods [35]. In order to know the sociodemographic characteristics and the most frequent chronic comorbidities among the study population, a descriptive analysis was carried out using frequencies, means and standard deviation.

For the clinical variables, the mean and standard deviation (SD) of each parameter at each of the study moments was calculated. If for the same individual there was more than one measurement for the same parameter, the mean and interquartile range (IQR) were calculated. To compare the differences in means in the clinical parameters between the previous measurement and the measurement at six months, the Student’s *t*-test for paired samples was used.

After carrying out the Levene tests of homogeneity of variances, the Student’s *t*-test (for equal or unequal variances) was used to compare the changes in clinical parameters between alcohol abuse disorder patients infected with COVID-19 and with those who did not contract the infection. For those variables with a fewer number of observations than 100, a Wilcoxon rank test was used.

Differences in drug consumption were assessed through the DHDs dispensed in the pharmacy to the study population in each of the periods. To determine the variations in the use of healthcare resources, the Student’s *t*-test for paired samples was also used.

A statistical analysis was carried out using IBM SPSS Statistic 21 (IBM Corporation, New York, NY, USA) and R version 4.0.5. (R Foundation for Statistical Computing, Vienna, Austria).

### 2.5. Ethical Considerations

This work was carried out under the principles of the Declaration of Helsinki and complies with the ethical standards of the Aragón Clinical Research Ethics Committee (study protocol PI20–175).

The Aragonese Health Service provided the medical records of the patients included in the study. The treatment, communication and transfer of these personal data was adjusted to the provisions of Regulation (EU) 2016/679 of the European Parliament and Organic Law on Protection of Personal Data and guarantee of digital rights March 2018.

## 3. Results

Six months before the start of the lockdown, 9576 men over 16 years of age in Aragon had a diagnosis of alcohol abuse in their EMR. 9184 (95.9%) did not become infected with COVID-19 during the study months; 392 (4.1%), however, presented with the infection during the same period of time. Of these, 39 cases were declared during the months of lockdown and 351 in the six months after its end. Regarding the number of deaths from other causes than COVID-19, three people died in the months prior to the declaration of the state of alarm, 41 during lockdown and 145 in the following six months.

The mean age of the sample was 56.4 (12.9). Almost two thirds (71.3%) had an income of less than EUR 18,000 per year and more than half (51%) lived in urban areas with more than 10,000 inhabitants. Among the male population with chronic alcohol consumption, the most frequent chronic comorbidities were: dyslipidemia (46.1%), followed by smoking (44.3%), hypertension (36.9%) and anxiety and depression (30.4%) [Table 1].

The variations in the clinical parameters of these patients when comparing the baseline measurement with the measurement in the following six months can be observed in Table 2. Only the subtle worsening observed in blood creatinine [*p* = 0.027 (95% CI: −0.06–−0.00)] and the slight improvement in LDL cholesterol [*p* = 0.043 (95% CI: 0.25–16.06)] presented statistically significant differences.

Table 3 shows the changes in drug use patterns among the population with this disorder. For all drugs used in the treatment of alcohol dependence, the total number of DHDs experienced a decrease in the six months following the end of lockdown. The same trend is shown in the total number of DHDs of antidepressants, with the exception of sertraline. In contrast, the total number of DHDs of benzodiazepines (diazepam, lorazepam, alprazolam, triazolam, lormetazepam, midazolam and loprazolam) was increased compared to the six months prior to the start of the lockdown.

Variations in the use of health resources by patients diagnosed with alcohol abuse can be seen in Table 4. In the six months after the end of the lockdown, the number of visits to the nurse at the health center, whether under ordinary [*p* < 0.001 (95% CI: 0.21–0.69)] or continued care [*p* = 0.005 (95% CI: 0.17–0.97)] experienced a statistically significant decrease. In contrast, the number of visits to the general practitioner at the health center for ordinary care [*p* < 0.001 (95% CI: −0.53;−0.23)] and the number of visits to social work services [*p* = 0.013 (95% CI: −3.14;−0.39)] showed a statistically significant increase in relation to the six months prior to the start of lockdown.

Regarding the performance of diagnostic tests on these patients, the number of X-rays, hemograms, biochemical, immunological and urine tests were reduced. In contrast, the number of resonances and microbiology tests increased [*p* < 0.005].

The number of visits to hospital care services (number of visits to accident and emergency services and number of hospitalizations) by patients diagnosed with alcohol abuse also shows a decrease which is only statistically significant for the number of hospitalizations [*p* < 0.001 (95% CI: 0.88–1.04)].

The comparison of the clinical parameters between patients with alcohol use disorder infected with COVID-19 and those not infected can be seen in detail in Table 5. Patients infected by COVID-19 presented higher BMI values than those who did not contract the infection. This difference was significant both for the six months prior to the start of lockdown [*p* = 0.010 (95% CI: −3.07; −0.42)] and for the six months after its completion [*p* = 0.001 (95% CI: −6.02;1.52)]. For the rest of the variables, no statistically significant differences were found.

## 4. Discussion

The results of our study do not show clinically relevant changes in the biological markers of this disease which would suggest the stable maintenance of alcohol consumption during the months of lockdown.

Other studies published in the population with alcohol use disorder also contradict the hypothesis of an increase in consumption during the health crisis when showing this maintenance [36], or even a decrease in the consumption [37,38] of this substance after the start of the sanitary measures established to stop the spread of the virus. However, some studies have reported an increase in alcohol consumption [24] and in the number of relapses during the months of lockdown [39,40] or having a higher risk of consumption those who experienced emotional distress, or lived alone at the onset of the pandemic [41]. Two studies carried out in our country [24] show opposite results as well. One of them analyzed the impact of COVID-19 in patients with alcohol use disorder collecting data during lockdown or immediately after the end of it. Its results showed results opposite to ours. However, our study considered the following six months, which could explain this opposite trend. The other study in Barcelona [36], which collected the data two months later to the end of lockdown, exposed a reduction in the frequency of alcohol use during lockdown. Nevertheless, this last study and others [36,37,39] only used self-reported

The maintenance of clinical parameters observed in our study could in part be explained by the continuous provision of care to these patients during and after lockdown. Some foundations in our community, such as the Zaragoza Solidarity Center—Proyecto Hombre (CSZ—PH), were recognized by the Ministry of Health of the Government of Aragon as essential services. This recognition allowed these services to remain open at all times, providing telematic or even face-to-face care to all patients and their families [42]. The same situation was found in other Spanish regions, where the maintenance of basic care for patients with alcohol use disorder seems to have cushioned the impact of COVID-19 by promoting abstinence and therefore protecting against possible relapses [43].

Regarding the variations in the consumption of drugs in the six months after the end of the lockdown, there are no notable differences in the total number of DHDs dispensed by the pharmacy in the drugs used for the treatment of dependencies or in the antidepressants. These slight variations would once again highlight the adequate management during the COVID-19 pandemic of the patient with alcohol abuse disorder [43] through telephonic consultations. On the other hand, the high rates of irritability, anxiety or somatization that several studies have reported [36,41] among this population during the months of lockdown could be behind the increase in the number of DHDs of some benzodiazepines (diazepam, lorazepam, alprazolam, triazolam, lormetazepam, midazolam and joprazolam).

The use of resources from the National Health System by patients with alcohol abuse disorder experienced a decrease compared to the six months prior to the start of lockdown. These results are in line with those published by the WHO in the survey on the impact of the pandemic on mental, neurological and substance use services [44], where 93% of the surveyed countries declared interruptions in these services. Telemedicine has been consolidated as one of the most frequent and effective alternatives to overcome these interruptions [43,44,45], and could explain the increase in the number of ordinary face-to-face or telephone visits to the general practitioners observed in our work. The increase, according to various reports published in our country [46,47], in the demand for social services in the months after the pandemic could explain the notable increase in the number of consultations to social work services.

Patients with substance abuse disorders usually present physical and mental comorbidities [28,29], which have been considered by some studies [48,49] as risk factors of greater severity of COVID-19 infection. The results of our work reflect a higher prevalence of obesity among patients with alcohol abuse disorder infected by COVID-19, which could contribute to a worse prognosis and evolution of the disease.

Regarding the excess of mortality by other causes observed in the six months after the end of lockdown, the emotional distress experienced [36,41] by this group during the pandemic months, the possible decompensations from other chronic comorbidities [50,51], together with the consequent economic crisis generated by COVID-19, could have increased mortality and the number of suicides [52]. Previous studies [53] have placed alcohol dependence among the diagnostic groups with the highest risk of suicide, behind depression and schizophrenia.

Our study has several limitations. First of all, we do not have access to a quantified record of alcohol consumption in Standard Drinking Units, nor to some of the specific structured questionnaires used in PC such as the CAGE or the Alcohol Use Disorders Identification Test (AUDIT). The exclusive use of the EMR also prevented the collection of self-reported data on the lifestyles maintained during the months of the pandemic. Second, we do not know the phase in which patients with alcohol abuse are at (detoxification, cessation or rehabilitation), therefore our results should be interpreted with caution. Thirdly, we did not consider the cause of death of these patients or the records collected during the months of strict lockdown, because these were very scarce. Nonetheless, lockdown could have had a particularly negative impact on alcohol consumption, as shown in a study in our country that was previously mentioned. Finally, the number of records for each of the clinical variables included in this study was very limited, so the power to detect significant differences is likely to be small. Access to these data depends on their validation by the general practitioner, so we only had access to the data that were validated by the doctor. Furthermore, the number of statistical tests and calculated *p*-values in this article is large and therefore needs to be confirmed in further studies.

## 5. Conclusions

Our study contributes to the knowledge from a longitudinal perspective of the consequences of the COVID-19 pandemic in the clinical, pharmacological and health resources use parameters of a large sample of men diagnosed with alcohol abuse disorder.

Although unforeseen, the results of our study suggest that the impact of COVID-19 among this group has been moderate. The reorganization of health and social services after the declaration of the state of emergency in our country made it possible to maintain care for this vulnerable group.

## Figures and Tables

**Table 1 ijerph-19-01358-t001:** Sociodemographic data and chronic comorbidities in alcohol abuse disorder patients from Aragon.

Age	Mean (SD)	56.4 (12.9)
		N (%)
Pharmaceutical service		
<18,000		6824 (71.3)
Between 18,000–100,000		2007 (21.0)
>100,000		19 (0.2)
Free pharmacy		696 (7.3)
Mutualist		26 (0.3)
Uninsured		4 (0.0)
Basic health area		
Urban		5279 (55.1)
Rural		4297 (44.9)
Chronic comorbidities (Yes %)		
Somatic comorbidities		7820 (81.7)
Arrhythmias		519 (5.4)
Heart failure		195 (2.0)
Ischemic heart disease		532 (5.6)
Hypertension		3534 (36.9)
Dyslipidemia		4418 (46.1)
Obesity		1327 (13.9)
Overweight		160 (1.7)
Disease in veins/arteries		505 (5.3)
Cerebrovascular disease		464 (4.8)
Diabetes		1584 (16.5)
Chronic bronchitis		182 (1.9)
COPD		1009 (10.5)
Asthma		356 (3.7)
Chronic kidney disease		337 (3.5)
Hypothyroidism		350 (3.7)
Hyperthyroidism		117 (1.2)
Anemia		947 (9.9)
Neoplasia		2015 (21.0)
Hearing loss		763 (8.0)
Cataracts		714 (7.5)
Glaucoma		406 (4.2)
Osteoarthritis		419 (4.4)
Osteoporosis		110 (1.1)
Dorsopathy		2197 (22.9)
Psychological comorbidities		6231 (65.1)
Smoking		4241 (44.3)
Insomnia		1377 (14.4)
Anxiety and depression		2910 (30.4)
Autolytic attempt		149 (1.6)
Dementia		140 (1.5)

Standard deviation (SD); Chronic obstructive pulmonary disease (COPD).

**Table 2 ijerph-19-01358-t002:** Changes in the clinical parameters of alcohol abuse disorder patients before and after lockdown.

		Six Months before	Six Months after	Difference between Pre—Follow Up	
	N	Mean (SD)		95% CI	*p*
GOT	87	33.5 (36.3)	29.7 (15.7)	−3.81; 11.31	0.327
GPT	94	29.5 (30.3)	27.6 (17.2)	−4.53; 8.31	0.561
Blood creatinine	110	0.9 (0.3)	1.0 (0.3)	−0.06; −0.00	0.027
Glomerular filtration	110	85.9 (20.9)	84.1 (21.4)	−0.08; 3.72	0.060
Systolic blood pressure	941	135.9 (15.8)	135.7 (16.5)	−0.83; 1.16	0.749
Diastolic blood pressure	941	78.0 (10.1)	77.9 (10.4)	−0.51; 0.70	0.752
Total cholesterol	109	204.5 (53.4)	198.3 (50.3)	−2.56; 14.93	0.164
LDL	92	119.9 (48.2)	111.7 (41.5)	0.25; 16.06	0.043
HDL	100	55.2 (19.4)	54.8 (19.0)	−2.16; 2.99	0.748
Triglycerides	102	148.3 (122.1)	151.2 (89.2)	−21.73; 15.91	0.760
BMI	347	30.3 (5.4)	30.2 (5.5)	−0.05; 0.27	0.179

Glutamate—oxaloacetate (GOT); Glutamate pyruvate (GPT); Low density lipoprotein (LDL); High density lipoprotein (HDL); Body mass index (BMI); Standard deviation (SD); Confidence interval (CI).

**Table 3 ijerph-19-01358-t003:** Number of DHDs six months before and six months after lockdown.

	Six Months before	Six Months after
Drugs used in addictive disorders		
Varenicline	4.22	2.87
Disulfiram	0.73	0.71
Acamprosate	0.001	0.001
Nalmefene	0.46	0.44
Naltrexone	0.17	0.14
Benzodiazepines		
Anxiolytics		
Diazepam	13.24	13.47
Potassium clorazepate	2.23	2.16
Lorazepam	61.97	68.91
Bromazepam	0.75	0.74
Clobazam	0.03	0.02
Ketazolam	0.24	0.24
Alprazolam	142.48	152.26
Hypnotics and sedatives		
Flurazepam	0.14	0.14
Triazolam	2.74	2.83
Lormetazepam	209.92	215.45
Midazolam	0.25	0.28
Brotizolam	1.77	0.73
Quazepam	0.01	0.01
Loprazolam	0.74	0.78
Antidepressants		
Non-selective monoamine reuptake inhibitors		
Imipramine	0.001	0.001
Clomipramine	0.09	0.08
Amitriptyline	0.11	0.10
Nortriptyline	0.003	0.002
Doxepin	0.00	0.00
Maprotiline	0.01	0.01
Selective serotonin reuptake inhibitors		
Fluoxetine	2.71	2.36
Citalopram	0.62	0.59
Paroxetine	2.81	2.80
Sertraline	1.37	1.41
Fluvoxamine	0.01	0.01
Escitalopram	17.15	16.65

Daily human dose (DHD).

**Table 4 ijerph-19-01358-t004:** Number of visits and diagnostic tests prescribed six months before and six months after lockdown.

			Six Months before	Six Months after		
		N	Mean (SD)		95% CI	*p*
No. of nursing visits (ordinary care) at health centre or by telephone		2738	4.72 (6.35)	4.27 (5.87)	0.21; 0.69	<0.001
No. of nursing visits (ordinary care) at home		135	6.10 (9.99)	6.63 (13.37)	−2.11; 1.04	0.505
No. of nursing visits (continuous care) at health centre		261	2.58 (3.49)	2.01 (2.08)	0.17; 0.97	0.005
No. of nursing visits (continuous care) at home		33	3.06 (7.31)	2.67 (6.60)	−0.50; 0.01	0.165 ^a^
No. of general practitioner visits (ordinary care) at health centre or by telephone		5799	5.19 (5.04)	5.57 (5.69)	−0.53; −0.23	<0.001
No. of general practitioner visit (ordinary care) at home		82	3.13 (2.98)	2.78 (2.50)	−0.53;0.01	0.345 ^a^
No. of general practitioner visits (continuous care) al health centre		523	2.03 (1.96)	2.12 (2.45)	−0.31; 0.12	0.395
No. of general practitioner visits (continuous care) at home		38	1.68 (0.93)	1.45 (1.06)	−0.49; 0.02	0.230 ^a^
No. of visits to other professionals						
	Social worker	56	2.70 (2.68)	4.46 (4.64)	0.01; 2.50	0.016 ^a^
No. of visits to specialised care (first consultation)		264	1.51 (0.94)	1.54 (1.00)	−0.19; 0.14	0.787
No. of visits to specialised care (successive consultations)		1812	2.72 (2.33)	2.71 (2.66)	−0.12; 0.13	0.925
No. of diagnostic test performed						
	X-rays	1335	1.23 (1.46)	1.07 (1.44)	0.07; 0.25	0.001
	Ultrasound	1335	0.36 (0.61)	0.32 (0.55)	−0.00; 0.07	0.079
	Resonance	1335	0.12 (0.36)	0.15 (0.41)	−0.06; −0.00	0.044
	CT scans	1335	0.46 (0.78)	0.51 (0.78)	−0.09; 0.01	0.101
	Digestive test	1335	0.01 (0.13)	0.01 (0.15)	−0.01; 0.01	0.777
	Hemograms	1178	0.34 (0.56)	0.28 (0.51)	0.03; 0.10	0.001
	Biochemistry	1178	1.04 (0.67)	0.85 (0.71)	0.14; 0.24	<0.001
	Microbiology	1178	0.20 (0.62)	0.33 (0.74)	−0.17; −0.07	<0.001
	Immunology test	1178	0.21 (0.45)	0.15 (0.40)	0.02; 0.08	<0.001
	Coagulation	1178	0.03 (0.17)	0.04 (0.22)	−0.02; 0.00	0.123
	Urine test	1178	0.31 (0.58)	0.24 (0.55)	0.03; 0.10	<0.001
No. of visits to A&E department		614	2.14 (2.51)	2.01 (2.29)	−0.04; 0.31	0.138
No. of hospital admission		666	1.37 (0.79)	0.41 (0.85)	0.88; 1.04	<0.001

Accident and emergency (A&E); Standard deviation (SD); Confidence interval (CI). ^a^ Wilcoxon signed- rank test.

**Table 5 ijerph-19-01358-t005:** Changes in the clinical parameters of patients with alcohol use disorder with COVID-19 and without COVID-19 before lockdown and six months after.

	Six Months before						Six Months after					
	With COVID		Without COVID				With COVID		Without COVID			
	N	Mean (SD)	N	Mean (SD)	95% CI	*p*	N	Mean (SD)	N	Mean (SD)	95% CI	*p*
GOT	32	30.3 (11.9)	615	34.8 (36.9)	−8.31; 17.39	0.488	35	35.5 (23.6)	460	38.0 (48.5)	−13.75; 18.78	0.761
GPT	35	34.5 (26.2)	656	33.0 (28.5)	−11.19; 8.16	0.758	36	38.6 (41.9)	496	35.4 (41.0)	−17.09; 10.75	0.655
Blood creatinine	37	0.9 (0.2)	700	0.9 (0.2)	−0.08; 0.07	0.829	36	0.9 (0.3)	544	1.0 (0.3)	−0.07; 0.13	0.512
Glomerular filtration	37	91.9 (18.1)	700	90.1 (18.0)	−7.75; 4.19	0.558	36	88.6 (17.2)	544	86.7 (20.0)	−8.67; 4.71	0.562
SBP	128	135.1 (17.7)	2519	135.5 (16.7)	−2.52; 3.44	0.761	60	134.3 (20.7)	1343	135.7 (17.1)	−3.09; 5.86	0.543
DBP	128	78.3 (10.9)	2515	79.6 (10.8)	−0.55; 3.28	0.162	60	77.4 (13.9)	1343	79.2 (11.2)	−1.09; 4.75	0.220
BMI	70	31.5 (6.0)	1277	29.7 (5.4)	−3.07; −0.42	0.010	24	33.6 (6.1)	585	29.8 (5.5)	−6.02; 1.52	0.001

Glutamate—oxaloacetate (GOT); Glutamate pyruvate (GPT); Systolic blood pressure (SBP); Diastolic blood pressure (DBP); Body mass index (BMI); Standard deviation (SD); Confidence interval (CI).

## Data Availability

Requests for any underlying data cannot be granted by the authors because the data was acquired under a license/data sharing agreement with the Aragon Health Services, under which conditions of use (and further use) apply.

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
