# Peer review of "Analysis of Clinical Parameters, Drug Consumption and Use of Health Resources in a Southern European Population with Alcohol Abuse Disorder during COVID-19 Pandemic"

_ijerph, 2022, doi:10.3390/ijerph19031358_

Round 1
Reviewer 1 Report
Many thanks to the Editor for allowing me to evaluate this work.
Addictions are still a significant health and social problem. Alcoholism, drug addiction and cigarette smoking are the most common, but behavioral addictions are also becoming more common. Experts emphasize that the increase in the number of addicts may be the hidden cost of the COVID-19 pandemic and related restrictions in social activity, isolation and anxiety for economic security. Experts indicate that the COVID-19 pandemic will significantly increase the health and social burden related to addiction . The methods of prevention and therapy should therefore undergo constant modification - as well as the definition and types of addictions. Therefore the manuscript deals with a very important social problem.
I found the paper very interesting, timely and scientifically sound. It is properly written, following all the guidelines for publications of scientific articles and it is within the scope of the International Journal of Environmental Research and Public Health. Overall evaluation: presented work contains 15 typewritten pages, including 5 tables, and 51 references thematically related to the content of the paper.
- the abstract is compendious
- the methods are properly described
- tables are presented correct
- discussion is extensive and exhaustive
- the manuscript contains a brief list of study' limitations and conclusions
I have only minimal comments:
In lines 90-93 the Authors claimed that there is a lack of studies on the impact of the pandemic and lockdown in the population with alcohol dependence. Recently, such studies have already been prepared. Please specify which articles it is about, perhaps there are no studies in Spain.
Line 189-191 the Authors should explain what was the cause of death of the indicated patients. Was it due to complications from alcoholism?
The Authors should explain why only patients with alcohol abuse disorder who did not contact COVID-19 were included since in the second part there were patients infected with COVID-19?
I agree that during the pandemic data were difficult to obtain (Line 113), however, they could have significantly influenced the results obtained in the later stages of the study. The Authors should comment this fact.
Line 342: since the Authors do not know what phase of addiction the patients were in, it is difficult to interpret the results. The situation of people at different stages of alcohol withdrawal is completely different.
I am also surprised by the presented results showing that the impact of COVID-19 on patients with alcohol abuse disorder was moderate. While most studies talk about a significant deterioration in the condition of such people. The Authors should comment on this data.
Reviewer 2 Report
Title: Analysis of clinical parameters, drug consumption and use of health resources in a Southern European population with alcohol abuse disorder during COVID-19 pandemic.
The study aims to analyze the changes in the clinical parameters, the consumption of drugs for the treatment of this addiction and the use of health resources, between the 6 months before the start of the lockdown and the 6 months after its end, in men diagnosed with alcohol abuse disorder in an autonomous community in northern Spain who did not contract COVID-19 infection during the study months; also comparing the variations in clinical parameters between uninfected alcoholic patients and those who were infected by the SARS-Cov-2.
Dear Authors,
The study is interesting and important for diagnostic and treatment of persons with alcohol use disorder in the pandemia and post-pandemic time, but authors should analyze and correct the manuscript contents:
1/ In the "Introduction": line 97 – Authors should explain the term „…new virus”. It is necessary because it is not clear.
2/Lines 74, 96, 237, 274, 279 – Please try to avoid using the word "alcoholic" and "alcoholism". Those word is becoming more stigmatized. The proposition of change of words e.g. patient with alcohol use disorder or patient with alcohol dependence.
3/ In the „Methods”: Authors should note the study type. Is the study a longitudinal or an analysis of independent groups in the two-time points? Because in the „Conclusion” the authors wrote -„Our study contributes to the knowledge from a longitudinal perspective of the consequences of the COVID-19 pandemic…”.
4/ In the "Methods": lines 156-174 – Authors can add the statistical programme name.
Next: Authors analyzed variables in the longitudinal study or not? It is important for use of statistical tests. The research concerns an analysis in tables no. 2-5.
Next: In the first part of table 2 is the comparison of three groups. What statistical programme was used?
Next: Are the authors sure of homogeneity of variances in the comparison of unequal groups? In table no. 5, e.g group with COVID-19 n=70 vs. group without n=1277?
5/ Line 200 – in Table 1 and the column „Chronic comorbidities (Yes %)” – I propose a set of results according to percentage and I propose dividing into parts “psychological and somatic”.
